# Fluorescent Auxin Analogs Report Two Auxin Binding Sites with Different Subcellular Distribution and Affinities: A Cue for Non-Transcriptional Auxin Signaling

**DOI:** 10.3390/ijms23158593

**Published:** 2022-08-02

**Authors:** Xiang Huang, Jan Maisch, Ken-Ichiro Hayashi, Peter Nick

**Affiliations:** 1Molecular Cell Biology, Botanical Institute, Karlsruhe Institute of Technology, Fritz-Haber-Weg 4, 76133 Karlsruhe, Germany; huangxiang@scbg.ac.cn (X.H.); jan.maisch@kit.edu (J.M.); 2Key Laboratory of South China Agricultural Plant Molecular Analysis and Genetic Improvement, Guangdong Provincial Key Laboratory of Applied Botany, South China Botanical Garden, Chinese Academy of Sciences, Guangzhou 510301, China; 3Department of Biochemistry, Okayama University of Science, 1-1 Ridai-cho, Okayama 700-0005, Japan; hayashi@dbc.ous.ac.jp

**Keywords:** auxin, indole acetic acid, binding site, NBD auxins, ER, tonoplast, auxin signaling

## Abstract

The complexity of auxin signaling is partially due to multiple auxin receptors that trigger differential signaling. To obtain insight into the subcellular localization of auxin-binding sites, we used fluorescent auxin analogs that can undergo transport but do not deploy auxin signaling. Using fluorescent probes for different subcellular compartments, we can show that the fluorescent analog of 1-naphthaleneacetic acid (NAA) associates with the endoplasmic reticulum (ER) and tonoplast, while the fluorescent analog of indole acetic acid (IAA) binds to the ER. The binding of the fluorescent NAA analog to the ER can be outcompeted by unlabeled NAA, which allows us to estimate the affinity of NAA for this binding site to be around 1 μM. The non-transportable auxin 2,4-dichlorophenoxyacetic acid (2,4-D) interferes with the binding site for the fluorescent NAA analog at the tonoplast but not with the binding site for the fluorescent IAA analog at the ER. We integrate these data into a working model, where the tonoplast hosts a binding site with a high affinity for 2,4-D, while the ER hosts a binding site with high affinity for NAA. Thus, the differential subcellular localization of binding sites reflects the differential signaling in response to these artificial auxins.

## 1. Introduction

Auxin acts as a major phytohormone during plant growth and development and is involved in almost every aspect of plant biology. The impact of auxin as a central developmental signal derives from its directional transport, which serves as an orienting cue to align and organize the entire architecture of a plant. This specific feature led to the discovery of this phytohormone: Charles Darwin and his son Francis [1] postulated a signal that, in phototropically stimulated grass coleoptiles, would convey the information about the direction of light from the irradiated tip to the non-irradiated base of the organ, where growth takes place. The search for this signal led Went (phototropism, [2]) and Cholodny (gravitropism, [3]) to independently propose a model of a transported growth hormone. Based on this model, auxin could be identified as indole acetic acid (IAA). The decisive criterion was its ability to evoke a coleoptile curvature in the *Avena* biotest [4,5]. Since then, our understanding of the molecular mechanism of auxin transport, as well as the activation of auxin-induced genes, has seen impressive advances.

Since auxin acts as an important information carrier, auxin itself must be under tight cellular regulation to avoid signaling chaos. Several mechanisms contribute to this control: de novo biosynthesis, conversion, storage (often in conjugated form (for review see [6,7])), oxidation, and degradation (reviewed in [8]), as well as directional import [9] and export [10]. The majority of auxin is synthetized in young and growing plant tissues. From there, it is transported to the target tissue, thus triggering both local [11]) and systemic [12] auxin responses. The pronounced directional movement through plant tissues makes auxin unique among the plant hormones. This peculiar phenomenon stimulated intensive research on polar auxin transport over the last five decades.

The classical model [13] assumes that indole-acetic acid, a very lipophilic molecule, is protonated in the acidic apoplast and, therefore, can permeate through the plasma membrane. However, inside the cytoplasm (pH 7.0), it will lose the proton, and the anionic IAA− form is trapped, such that export relies on active auxin efflux carriers, such as the PIN FORMED (PIN) or the ATP-BINDING CASSETTE GROUP B (ABCB/MDRPGP) proteins [14,15]. The polar localization of these efflux carriers can explain the distinct directionality of auxin transport and the formation of auxin gradients across a tissue [16,17,18]. For the synthetic auxin analog 1-naphthaleneacetic acid (NAA), which is much more hydrophilic than IAA, import does not occur through this ion-trap mechanism, but by auxin influx carriers [19] (reviewed in [20]) that are localized in a polar fashion at the cell pole opposite to the PIN efflux carriers [21]. Thus, directional influx can also contribute to the polarity of the auxin flow. However, to fulfill its function in the self-organization of vascular bundles, the non-directional influx produced by the ion-trap mechanism proposed by the classical model is essential (discussed in [22]).

Although auxin is a central signal molecule for numerous and quite different responses, its structure is quite simple. Thus, the ability of auxin to instruct numerous cells in a specific manner cannot derive from the molecule itself but must derive from the perceiving cellular system. The molecular mechanism responsible for the modulation of the gene expression has become relatively clear (reviewed in [23]). In brief, auxin triggers transcriptional responses via the auxin receptor TRANSPORT INHIBITOR RESPONSE1/AUXIN SIGNALING F-BOX (TIR1/AFB). Upon the binding of auxin, the repressor proteins of the Auxin/INDOLE-3-ACETIC ACID (Aux/IAA) family are doomed to proteolysis, which will release AUXIN RESPONSE FACTORS (ARFs) that can then regulate different target genes.

There is no doubt that the interaction between Aux/IAAs and TIR1/AFB is important to auxin biology, modulating the transcription of auxin-responsive genes. However, other auxin signaling systems that complement this canonical pathway also exist. For instance, the auxin-dependent cleavage of the membrane-located receptor kinase TRANSMEMBRANE KINASE 1 (TMK1) releases a fragment that interacts with INDOLE-3-ACETIC ACID 32 and INDOLE-3-ACETIC ACID 34 in the nucleus [24]. To what extent TMK1 is an auxin receptor is not clear, but it clearly triggers a parallel pathway initiating from the plasma membrane.

We should keep in mind that not all plant responses run over transcriptional activation. Some plant growth responses to auxin initiate very rapidly, which seems unlikely, due to transcriptional regulation [25]. For instance, in the root curvature of *Arabidopsis thaliana*, the auxin efflux from the root cap to the elongating cells triggers the accumulation of cytosolic Ca^2+^ within 7–14 s, concomitantly with apoplastic alkalinization [26]. As the auxin receptor for such rapid responses, complementing the activity of the TIR1/AFB signaling system, AUXIN BINDING PROTEIN1 (ABP1) has been intensively studied for decades (for a classical review see [27]; updates are given in [28]). However, loss-of-function mutants for ABP1 in thale cress show normal growth and development, casting doubt on the importance of ABP1 for auxin signaling [29]. However, a very careful and meticulous analysis using different auxin analogs and mutants later rehabilitated the importance of ABP1 for auxin-induced rapid protoplast swelling; however, the analysis confirmed that this putative receptor was dispensable for auxin-induced hypocotyl growth [30].

Thus, it is clear that more than one receptor exists, and that different signal pathways are deployed. If this holds true, different auxins should, therefore, modulate different physiological processes with different dose response relations. Using cell division and cell expansion as the readout in tobacco suspension cells, we showed in our previous work [31] that the artificial auxin NAA preferentially stimulated cell expansion. This indicated that the two artificial auxins trigger different signaling chains. In fact, the effect of 2,4-dichlorophenoxyacetic acid (2,4-D) on cell division could be mimicked by aluminum tetrafluoride, an activator of trimeric G-proteins, and it could be disrupted by the pertussis toxin, a specific inhibitor of trimeric G-protein signaling. In contrast, NAA-induced cell expansion proceeded independently of trimeric G-protein signaling.

Insight into the subcellular localization of auxin might help to understand how auxin can modulate cellular responses via different signaling pathways. Several methods allow us to track auxin within a tissue. For instance, auxin-inducible genes fused with reporter constructs [32,33], antibodies against auxin or its carriers [34,35], or mass-spec-based measurements [36,37] have been successfully employed to detect or at least infer the auxin content in plant tissues. Each technique has its advantages and disadvantages; some are indirect, such as the approach to visualize auxin-induced promoter activities, while others are destructive, such as that it is not possible to follow fluctuations over time. None of them allow resolving the subcellular patterns. This also holds true for the most recent of these sensors, based on a bacterial tryptophan repressor that has been engineered, such that a fluorescent signal is produced upon the binding of auxin through fluorescence resonance energy transfer [38]. Although this sensor excels its predecessors by the fact that the signal is switched off, once the auxin disappears, such that temporal dynamics can be followed, it only allows for a scalar readout of a given cell. The signal just indicate that auxin is present in this cell, not where it is in the cell. As an alternative approach to genetically encoded sensors, auxin analogs that mimic native auxin with respect to transport allow us to increase resolution beyond the cell level. In fact, using radioactively labelled azido-IAA, the repartitioning of auxin between different tissues of etiolated corn mesocotyls in response to a red light became visible [39]. However, the samples must undergo chemical fixation, which means that the method does not allow for studies in vivo. Fluorescently labeled auxin analogs allow us to overcome this drawback [40]. These analogs retain their activity with respect to the auxin transport but are inactive with respect to the auxin signaling and metabolism and, thus, should not perturb auxin homeostasis. This approach allows us to trace auxin distribution in vivo with unprecedented resolution. Of course, this strategy implies the precondition that the analogs bind to the same receptors as innate auxin and, thus, truly reflect the behavior of the innate auxins they supposedly represent.

In the current study, we analyzed the subcellular distribution of fluorescent auxin analogs. We found that different auxin species bind to different subcellular targets (ER and tonoplast). Since (unlabeled) auxins can outcompete the analogs from the binding sites, the patterns that manifest from these fluorescent probes are specific and report the pattern of the auxin species they stand for. These patterns bear upon current models of auxin transport and signaling.

## 2. Results

### 2.1. The Fluorescent Auxin-Analogs NBD-NAA and NBD-IAA Localize Differently

To investigate the subcellular distribution of fluorescent auxin analogs, non-transformed tobacco BY-2 cells were incubated with either 7-nitro-2,1,3-benzoxadiazole (NBD)-NAA (2 μM) or NBD-IAA (2 μM) for different time intervals. Subsequently, we washed out the unbound compounds. Auxins can cross the plasma membrane to enter the cytoplasm either by an ion-trap mechanism or by influx carriers. Therefore, we selected one very early (1 min) and one later (20 min) time point for the time course. When NBD-NAA was visualized after 1 min of incubation (Figure 1C), a punctate pattern was observed. This signal was not only aligned with the cell membrane, but occasionally also seen in the trans-vacuolar strands (Figure 1C). For prolonged incubation (20 min), instead of these dots, a continuous membrane-like signal lined the trans-vacuolar strands (Figure 1D–F), probably representing the tonoplast. The pattern obtained for NBD-IAA was initially similar with a dot-like distribution (Figure 1I). However, in contrast to NBD-NAA, the NBD-IAA signal remained punctate throughout (Figure 1L).

Thus, the subcellular distribution patterns of NBD-NAA and NBD-IAA differentiated with time, whereby NBD-NAA shifted to the tonoplast, while NBD-IAA remained in the punctate structures appearing immediately after uptake.

### 2.2. NBD-NAA Links to Endoplasmic Reticulum (ER) and Tonoplast, NBD-IAA to the ER

As mentioned above, the distribution patterns of fluorescent auxin analogs are different when the incubation time is sufficient to establish the final localization (see Figure 1F,L). So as to understand the cellular base for the localization of NBD-NAA and NBD-IAA, we used several fluorescent markers to test their co-localization with NBD-NAA or NBD-IAA, respectively. 

To examine whether the fluorescent auxin analogs were localized to the ER, non-transformed BY-2 cells were incubated with NBD-NAA (2 μM) and ER-Tracker-Red (1 μM). Both signals overlapped in the trans-vacuolar strands of the perinuclear region, as well as close to the plasma membrane, indicative of co-localization in these areas (Figure 2C). However, in some areas, both signals were clearly separate. This included broad trans-vacuolar strands (Figure 2C, inset), where the ER-Tracker signal was found in the center of the strand, while the NBD-NAA signal was lining the interface with the vacuole or formed filamentous structures running across the strand. In some cases (Figure 2F), the two signals deviated almost completely, whereby the NBD-NAA signal showed the smooth lining typical of vacuolar membranes. The patterns seen for NBD-IAA (2 μM) and ER-Tracker-Red (1 μM) differed significantly. Here, the two signals co-localized strictly (Figure 2I,L). The double staining with ER-Tracker Red indicates that NBD-IAA links tightly with the ER, while NBD-NAA, in addition, also localized to a different structure, which might be the tonoplast.

To elucidate whether the second structure labelled by NBD-NAA represents the tonoplast, BY-2 cells were transiently transfected with the gene for tobacco TWO PORE CALCIUM (NtTPC1A) in fusion with RFP. This marker encodes a vacuolar calcium channel [40,41] and, therefore, highlights the tonoplast. In fact, the signal for NBD-NAA (2 μM) was clearly congruent with the NtTPC1A-RFP signal (Figure 3C,F), supporting the notion that NBD-NAA associates with the tonoplast. So as to delineate this putative tonoplast pattern from a potential association with the plasma membrane, we generated protoplasts from BY-2 cells by digesting the cell wall with cellulose and pectolyase. When the cell wall is removed, the turgor pressure will turn the protoplast into a spherical shape, such that the plasma membrane can be clearly separated from the tonoplast, because the vacuole assumes a reticulate pattern under these conditions (Figure 3I,J) and so does the distribution pattern for NBD-NAA (Figure 3G,H).

For comparison, we assessed the behavior of PIN1-GFP as a plasma membrane marker under the same conditions. This marker was present as a punctate signal in or at least very close to the plasma membrane (Figure 3K,L) but did not produce the long filamentous structures seen for NtTPC1A-GFP and for NBD-NAA. Thus, the dual visualization with a tonoplast marker, as well as the comparison of the patterns with tonoplast and plasma membrane markers in protoplasts implied that NBD-NAA is associated with the tonoplast.

### 2.3. NAA Can Compete with NAA-NBD for the Same Binding Sites

The specific subcellular localization of the fluorescent auxin analogs (see Figure 2 and Figure 3) indicates that they are interacting with specific binding sites. If they shared these binding sites with the non-fluorescent auxins, one would expect that the signal could be outcompeted by an excess of non-labelled auxins. Therefore, it was systematically tested whether NBD-NAA or NBD-IAA can compete with different auxins (NAA, IAA, and 2,4-D) for the same binding sites. If so, changes in the fluorescent signal of NBD-NAA or NBD-IAA should reflect the binding characteristics of the respective type of auxin (which is invisible).

To investigate whether NBD-NAA and NAA share the same binding sites and whether this binding process was reversible or not, non-transformed BY-2 cells were first incubated with different concentrations of NAA (2–100 μM) for 20 min before adding NBD-NAA (2 μM) and incubating for another 20 min (Figure 4A–D). In a second set of experiments, the cells were treated with NBD-NAA (2 μM) and, simultaneously, with different concentrations of unlabeled NAA for 20 min (Figure 4E–H). The results for both treatments were identical. This shows that the bound NBD-NAA can be fully outcompeted by an excess of non-labelled NAA and thus, just follows the chemical-mass equilibrium. In other words, the interaction between NAA and the binding site is fully reversible. While 2 μM of non-labelled NAA did not alter the pattern seen for NBD-NAA (compare Figure 4A,B,E,F to Figure 1D–F), the signal progressively faded when the concentration of NAA increased. For the highest tested concentration (100 μM, Figure 4C,D,G,H), only a faint residual fluorescent signal was detectable. These findings suggest that NBD-NAA and NAA compete for the same binding sites, and that binding is a reversible process.

To further test whether other auxins (IAA or 2,4-D) would also compete with NBD-NAA, non-transformed BY-2 cells were co-incubated with NBD-NAA (2 μM) and either IAA (Figure 5A–D) or 2,4-D (Figure 5E–H) for 20 min, again in several concentrations. Exemplarily, the lowest (2 μM, Figure 5A,B for IAA, Figure 5E,F for 2,4-D) and the highest (100 μM, Figure 5C,D for IAA, Figure 5G,H for 2,4-D) concentrations are shown. In contrast to NAA (see Figure 4), none of these auxins (IAA or 2,4-D) were able to quench the fluorescent signal from NBD-NAA, even if given in a 50-fold excess (100 μM) over NBD-NAA (2 μM). (Figure 5A–H). However, 2,4-D, administered in this excessive concentration, altered the distribution pattern of NBD-NAA from the usual tonoplast-like pattern to a vesicular pattern (Figure 5H). To find out whether this altered pattern for the distribution of NBD-NAA was possibly caused by the disintegration of the vacuole into vesicles, the tonoplast-marker strain NtTPC1-GFP was treated with different concentrations of 2,4-D. The results obtained for the lowest (2 μM) and the highest (100 μM) concentration of 2,4-D are exemplarily shown in Figure 5I–L. While the tonoplast-maintained integrity under treatment with 2 μM 2,4-D (Figure 5I,J), it appeared to disintegrate into vesicular structures for the highest (100 μM) concentration of 2,4-D (Figure 5K,L). These results imply that neither IAA, nor 2,4-D can compete with NBD-NAA for the same binding sites. The changed localization of NBD-NAA seen under 100 μM of 2,4-D is correlated with a loss of tonoplast structure produced by these excessive concentrations of 2,4-D.

Using the same experimental design, we tested whether NBD-IAA can compete with the different auxins (IAA, NAA, and 2,4-D) for their binding sites (Appendix A) using non-transformed BY-2 cells. The competition of NBD-IAA with its cognate auxin, IAA, was not as evident as in the case of NAA (compare Appendix A for NBD-IAA and the respective images in Figure 5A–D for NBD-NAA). Here, the excess of unlabeled IAA (100 μM) did not significantly quench the overall fluorescence. However, a closer look reveals that the bright punctate signals seen for a low concentration of unlabeled IAA (2 μM, Appendix A) were hidden by adding an excess of unlabeled IAA (Appendix A), while the diffuse background in cytoplasmic strands persisted. This is in clear contrast to the tonoplast signal seen with NBD-NAA, where the unlabeled NAA clearly quenched the fluorescent signal (Figure 5C,D). In contrast to non-labelled IAA, neither the non-labelled NAA (Appendix A), nor the non-labelled 2,4-D (Appendix A) produced any significant quenching of the punctate signals produced from the NBD-IAA.

In summary, these competition experiments show that both labeled auxins can be outcompeted by their unlabeled cognate but not by other auxins. There is, however, a difference with respect to the subcellular localization in response to excessive (100 μM) concentrations of 2,4-D; the fragmentation of the vacuole caused by this treatment (Figure 5K,L) will alter the localization of NBD-NAA (Figure 4G,H), while the localization of NBD-IAA remains unaltered. The ER-associated pool of NBD-NAA responds differentially to IAA and 2,4-D

In contrast to NBD-IAA, we observed two subcellular pools for NBD-NAA, one associated with the ER, the other with the tonoplast. While the addition of IAA and 2,4-D did not reduce the signal of NBD-NAA, they still might modulate the subcellular distribution of this binding site (see Figure 4 and Figure 5). To analytically probe for such potential modulations, we quantified the co-localization of NBD-NAA with the ER-tracker and NBD-NAA under low (2 μM) and high (100 μM) concentrations of unlabeled IAA and 2,4-D (Figure 6). In the absence of IAA or 2,4-D, the co-localization coefficients for NBD-NAA (giving the proportion of total NBD-NAA which was associated with the ER) were around 35–45% (left black columns, Figure 6). In contrast, the respective values for the ER-Tracker were close to 100% (right black columns, Figure 6), meaning that the proportion of the ER which was decorated with NBD-NAA was close to unity. Thus, the binding sites on the ER for NBD-NAA were limiting and could recruit only a part of the NBD-NAA population.

When we added a low concentration (2 μM) of IAA, this reduced the coefficient for NBD-NAA significantly by around 25% (Figure 6A). This reduction was accentuated to around 50% with 100 μM IAA (Figure 6B), which is indicative of a release of NBD-NAA from the ER (which remained fully decorated with NBD-NAA, as indicated by a coefficient for ER-tracker that stayed constantly close to 100%). Thus, the limitation in the ER-located binding sites for NBD-NAA was amplified depending on the concentration of exogenous IAA. When this experiment was repeated with 2,4-D, the pattern was different (Figure 6C,D). For a low concentration of 2,4-D, the correlation coefficient for NBD-NAA remained unchanged as compared to the solvent control (Figure 6C). In contrast, 100 μM 2,4-D significantly enhanced the weighted co-localization coefficients of NBD-NAA (Figure 6D) by around 25%. This indicates that the excess of 2,4-D shifted the NBD-NAA towards the ER.

### 2.4. Quantification of the Competition between NBD-NAA and Unlabeled NAA

As NBD-NAA and NAA obviously competed for the same binding sites (see Figure 4), a dose–response curve for this competition was constructed by applying a constant concentration of NBD-NAA (2 μM) in combination with progressively increasing concentrations of NAA. Then, we measured the fluorescent intensity of the remaining NBD-NAA signal by quantitative image analysis to estimate the degree of competition. With increasing concentrations of unlabeled NAA, the relative fluorescence intensity (estimated as the corrected single cell fluorescence, CSCF) decreased progressively (Figure 7A). The resulting quenching effect (Figure 7B) was then fitted using a Michaelis–Menten model (i.e., assuming steady-state equilibria), reaching optimal congruence with the measured data with an estimated affinity constant of 1.15 μM NAA, which is consistent with the physiological data on NAA effects on tobacco BY-2 cells [42].

## 3. Discussion

In the current study, we used fluorescent analogs of auxin that are not able to trigger signaling but migrate as their unlabeled counterparts, such that their subcellular distribution should reflect the behavior of innate auxin. We observe that NBD-NAA decorates the endoplasmic reticulum and tonoplast, while NBD-IAA binds to the ER only. The binding site for NBD-NAA at the ER can be outcompeted by unlabeled NAA, but not by IAA. This competition experiment allows us to estimate the affinity (using a Michaelis–Menten) to be around 1 μM, which is in the physiological range and further underpins specificity of these binding sites. In the following, we will first discuss to what extent these fluorescent analogs report the behavior of endogenous IAA and non-labeled NAA. Then, we develop a working model that uses two binding sites (at the ER and at the tonoplast) that differ in their affinities for IAA and NAA and integrate this model with empirical evidence for auxin signaling. This requires a new perspective regarding auxin transport leading to implications that are testable in future work.

### 3.1. Do Fluorescent Auxin Analogs Report the Subcellular Distribution of Auxin?

Visualization with fluorescent markers has allowed us to understand cells as dynamic processes rather than as static objects. This new viewpoint comes with a price, though. To visualize something means to change it, the label might block a moiety of the labeled molecule that otherwise would mediate interactions with other molecules, and this might interfere with subcellular localization or with innate dynamics. On the other hand, label-free visualization is limited by the diffraction limit and by the inability to distinguish the object of interest from other molecules or structures. This conundrum represents something similar to the biological version of Heisenberg’s uncertainty principle, describing the impossibility of determining both the location and movement of a particle at the same time [43]. For a small molecule such as auxin, the difficulty to follow its location seems comparable. Given the very small size of the label, the NBD-conjugated auxin analogs might be the best approximations currently available, since they phenocopy the transport properties of their unlabeled counterparts but cannot deploy auxin signaling such that they should not perturb auxin homeostasis [40]. We now observe a different subcellular localization of NBD-IAA and NBD-NAA, leading to the question: to what extent do these patterns reflect a differential subcellular localization of IAA and NAA? Since IAA and NAA themselves are not visible, this task resembles an equation with two variables. However, if details about these observed patterns match implications on subcellular localization inferred from the physiological effects of IAA and NAA, this might serve to provide conjecture on homologies.

What did we learn regarding the behavior of NBD-IAA and NBD-NAA, and to what extent can we link this to what we know about the physiological effects of their unlabeled counterparts?

Although NAA often serves as a stable chemical analog for the natural auxin IAA, it differs not only with respect to transport, but also with respect to biological activity. For instance, while the uncharged form of IAA can permeate from the acidic apoplast through the plasma membrane into the cytoplasm, where it loses its proton and remains trapped [13], the more hydrophilic NAA requires the AUX1 importer to enter the cell [19]. The differences in transport behavior also reflect the differences in the physiological effect. Cell elongation is under the control of a receptor system with a high affinity for NAA, while cell division requires higher concentrations of NAA [44]. Both auxin effects seem to differ in their signaling as well, since cell division requires the activation of a trimeric G-protein [45]. In fact, in the tobacco cell line VBI-0, where cell division and cell expansion separate more distinctly in time than in the BY-2 system, we showed that cell proliferation was more sensitive to activation by 2,4-D, while cell expansion was more sensitive to activation by NAA [31]. Furthermore, the response to 2,4-D could be blocked by the G-protein inhibitor pertussis toxin and activated by the G-protein activator aluminum tetrafluoride, while the response to NAA was not affected. Thus, there is compelling evidence for two auxin receptors (or, more precisely, for two perceptive events) that differ in their affinities for different ligands, deploy different signaling, and activate different cellular responses.

Different receptors (or perceptive events) activating different cellular responses might locate to different sites in the cell. While this conclusion is not necessarily true, there are three arguments that support the notion of two intracellular sites (perceptive events) for auxin binding. They are discussed below.

The first argument comes from a comparative study on ligand specificities for the auxin transport and the auxin activation of growth across a wide range of plant species in both shoots and roots [46,47]. They observe that the effect of the analog 2-naphthoxyacetic acid (2-NAO) on growth correlates with its ability for polar transport. Both are lost in Poaceae coleoptiles, contrasting with all the other systems. The loss of the ability to transport and respond to 2-NAO, thus, represents a synapomorphy of the Poaceae in sensu Hennig [48]. The straightforward explanation for a synapomorphy of two characters assumes that they depend on the very same mutation. Thus, using Occam’s razor, it is fair to conclude that the transport and perception of auxin seem to depend on the same protein.

At this point, the second argument kicks in. If the binding of NBD-IAA to the ER reflects the localization of an innate auxin receptor, this receptor might also be an auxin transporter. Canonical auxin transporters are usually located in the plasma membrane, at least in their functional form. The fact that they undergo recycling and, thus, can pass through an intracellular state as well [49] does not impair this statement, because, so far, there is no evidence that this transient passage form is transport competent. Moreover, the BFA compartments consist of a trans-Golgi network, and Golgi are clearly distinct from the ER [50]. This reasoning leads us to the question of whether a functional auxin transporter acts at the ER. In fact, this seems to be the case. PIN5, an atypical member of the PIN family, tethers to the ER [51]. Upon overexpression, it can retain auxins in both the heterologous yeast as well as the homologous *Arabidopsis thaliana* protoplast system indicative of transporter activity. However, these findings would also be compatible with a role of PIN5 as binding site for auxin (both interpretations are not mutually exclusive, though).

The third argument derives from the current study. While NBD-NAA first binds to the ER and later shifts to the tonoplast, NBD-IAA remains bound to the ER throughout. We have used two fluorescent organelle markers, ER-Tracker Red (Figure 2) and TPC1a-RFP (Figure 3), to assess the localization of the NBD auxins, and we have verified the tonoplast localization by a third marker, PIN1-GFP, to confirm that the TPC1a-marker labels the tonoplast and not the plasma membrane (Figure 3K,L). Unlabeled NAA can prevent NBD-NAA from binding to the ER with an estimated affinity of around 1 μM, which is well in line with the physiological effect. The quenching works independently of the temporal sequence (simultaneous application versus preloading with NAA and subsequent addition of NBD-NAA), meaning that it is not a competition for uptake but a competition for intracellular binding. An unbound signal will be diffuse and, therefore, mostly filtered out from confocal sections as non-structured background. In contrast to unlabeled NAA, unlabeled IAA is less effective but still can reduce the co-localization of NBD-NAA with the fluorescent ER-tracker. Thus, the ER harbors a binding site with a high affinity for NAA and for IAA, while the tonoplast harbors a binding site with a still-substantial affinity for NAA and a low affinity for IAA. Since cell expansion is more sensitive to NAA [31,43], the binding site at the tonoplast might be responsible for cell expansion, while the binding site at the ER might be responsible for cell division.

The temporal shift of NBD-NAA from the ER to the tonoplast might derive from the dissociation of the binding site at the ER and the rebinding to the binding site at the tonoplast. However, it might also derive from the transfer of the binding site at the ER to the tonoplast via the vesicle flow through Golgi, trans-Golgi network, and multivesicular body to the tonoplast [52,53,54]. If so, the blocking of vesicle flow should retain NBD-NAA at the ER, preventing the tonoplast localization. Phytotropins, such as 1-Naphthylphthalamic acid, or 2,3,5-triiodobenzoic acid (TIBA), cause the bundling of actin [55,56] and, thus, disrupt the vesicle flow. In fact, NBD-NAA co-localized with the ER-tracker in tobacco BY-2 cells pre-treated with phytotropins without showing a tonoplast signal [40]. Likewise, in differentiated root-cortex cells, as well as in the root hairs of *Arabidopsis thaliana*, the signal was congruent with CFP fused to the ER-retention motif HDEL and did not associate with the tonoplast. These cells harbor a fully expanded vacuole, such that the intensity of the vesicle flow from the ER to the tonoplast is much lower than in the proliferating tobacco BY-2 cells. Our observations concur with those findings supporting the notion that the binding site of NBD-NAA, not NBD-NAA itself, repartitions between the ER and the tonoplast. The binding site for NBD-IAA remains tethered to the ER, which is further evidence of two separate perceptive events.

### 3.2. Two Perceptive Events, Two Types of Auxin Transporters?

In this context, it is interesting that PIN5, in contrast to PIN1, which only accepts IAA as cargo, also transports NAA [50], which is consistent with a transporter–receptor model. The reason for the specificity might be the structural difference between IAA (indole ring) and NAA (naphthalene ring). The TIR1 receptor shows a similar differential affinity for these two auxins [57], and the same holds true for several influx carriers [20,58]. With respect to the PIN efflux carriers, all can transport IAA, but only few accept NAA [59,60].

Of course, one has to keep in mind that a labelled analog is always just a model for its native template, and its behavior will depend on the type of label. For instance, fluorescein-isothiocyanate or rhodamine-isothiocyanate conjugates of IAA retained IAA-like activities in roots of Arabidopsis [61], while NBD-IAA turned out to be completely inactive with respect to the signaling and metabolism in the same system [40].

The PIN transporters able to transport NAA and localize to the ER are of the “short” type [51,59,62], which might also be the reason for their lower selectivity. In contrast, the “long” PIN proteins (PIN1-4 and PIN7) show polar plasma membrane-localization and do not transport NAA, but IAA [58,63,64]. Neither NBD-NAA nor NBD-IAA exhibited any signal at the plasma membrane speaking against the possibility that the “long” PIN proteins act as binding sites. However, the ER localization of both NBD-NAA, and NBD-IAA would be consistent with a model where the “short” PIN proteins mediate the binding.

While many aspects of the two binding sites becoming manifest through labeling with fluorescent auxin analogs remains unknown, it is possible to develop a working model that should help to structure future research (Figure 8). The core of this model consists of two binding sites for auxin that are located on the ER and differ in their ligand patterns and in their responses to ligand binding. One binding site has a high affinity for IAA and, upon binding, remains at the ER. Its function might be the stimulation of cell division. The other binding site has a high affinity for NAA and, upon binding, repartitions to the tonoplast. Its function might be the stimulation of cell expansion. Since NBD-auxins are not physiologically active, it is not possible to address their physiological role directly. However, based on dose–response curves with different auxins in tobacco cells, it can be shown that cell expansion is more sensitive to NAA, while cell division is not [31]. The mechanism for this repartitioning event still awaits clarification, but the most straightforward mechanism would be the assignment of the binding site to vesicles that pass via the Golgi and multivesicular body to the vacuole [52]. Whether these binding sites overlap or are identical with “short” PIN proteins would represent a second topic for future research. 

The sequestration of auxin to membrane-tethered binding sites bears on the role of auxin for cellular directionality. While gradients from freely diffusible auxin would dissipate over time and, thus, would require continuous re-enforcement, a gradient of tethered auxin would remain stable over a longer time, which would facilitate the stable polarization of a cell. Polarization by intracellular auxin gradients seems to occur, for instance, in the positioning of root hairs (for a discussion, see [65]). How the binding of a ligand to a receptor turns on a signal is a classical topic in biology. To turn a signal off again often attracts a lower degree of attention. It is relevant, though—the role of “short” PIN proteins might consist of the sequestration of auxin, which would dissipate the signal. Alternatively, the translocation of a signal to a different site in the cell through vesicle traffic would allow for signal dynamics. For instance, the NAA-binding site might first deploy a signal at the ER, and the migration to the tonoplast would then be either a part of the switch-off (similar to the endocytosis of the flg22 receptor [66]), or it might be the cellular manifestation of a second signaling event with a different functional context.

## 4. Materials and Methods

### 4.1. Tobacco Cell Cultivation

BY-2 (*Nicotiana tabacum* L. cv Bright Yellow 2) suspension cell lines [67] were cultivated in a liquid medium containing 4.3 g/L Murashige and Skoog salts (Duchefa, Haarlem, The Netherlands), 30 g/L sucrose, 200 mg/L KH_2_PO_4_, 100 mg/L inositol, 1 mg/L thiamine, and 0.2 mg/L (0.9 μM) of 2,4-D, adjusted to pH 5.8. The cells were sub-cultured weekly, inoculating 1.0–1.5 mL of stationary cells into fresh medium (30 mL) in 100-mL Erlenmeyer flasks. The cells developed at 26 °C under constant shaking on a KS260 basic orbital shaker (IKA Labortechnik, Staufen, Germany) at 150 rpm. Every three weeks, we sub-cultured the stock of BY-2 calli on media solidified with 0.8% (*w*/*v*) agar (Roth, Karlsruhe, Germany). The cells and calli of the transgenic BY-2 strain NtTPC1A-GFP, stably transformed BY-2 cells with an NtTPC1A (*Nicotiana tabacum* Two Pore Channel 1A)-GFP construct [41] were cultivated on the same media as non-transformed wild-type cultures (BY-2 WT) but supplemented with 100 mg/L Kanamycin. The authors acknowledge Dr. Q. Liu (Botanical Institute, Karlsruhe Institute of Technology) for kindly providing the strains NtTPC1A-GFP and -RFP [68]. The cells and calli of the transgenic BY-2 strain PIN1-GFP, stably transformed BY-2 cells with a fusion construct of PIN1 (pin-formed protein 1, from *Arabidopsis thaliana*) and GFP [10], were cultivated in the same media as the BY-2 WT cultures but supplemented with 40 mg/L Hygromycin. The authors acknowledge the provision of this strain by Dr. J. Petrášek (Institute of Experimental Botany, Academy of Sciences of the Czech Republic, Prague, Czech Republic).

### 4.2. Fluorescent Auxin Analogs

Two fluorescent auxin analogs, NBD-NAA (7-nitro-2,1,3-benzoxadiazole conjugated to NAA) and NBD-IAA [40] were used in this study. These analogs closely mimic NAA and IAA with respect to polar transport but are inactive for auxin signaling and metabolism. Therefore, they allow us to visualize auxin transport and distribution without disturbing the auxin-signaling pathway. Both analogs were dissolved in dimethylsulfoxide (DMSO; Carl Roth GmbH, Karlsruhe, Germany) to yield a stock solution of 0.5 mM, which was frozen in aliquots in a deep freezer (−80 °C). From preparatory tests, a concentration of 2 μM was selected as the final experimental concentration.

### 4.3. Treatment with Fluorescent Auxin Analogs

If not stated otherwise, 1 mL of non-transformed (WT) cells, collected one day after sub-cultivation, was incubated with the fluorescent auxin analogs (NBD-NAA or NBD-IAA) in a final concentration of 2 μM for 20 min on an orbital shaker (IKA-WERK, Staufen, Germany) at slow rotation (20 rpm). We used custom-made staining chambers using mesh with a pore-size of 70 μm as bottom [69] to drain off the excess medium and to wash twice with fresh medium to remove unbound fluorescent auxin analogs.

To relate the subcellular localization of the fluorescent auxin analogs to the ER, we pre-incubated with fluorescent auxin analogs, as mentioned above. Subsequently, we added ER-Tracker (ER-Tracker™ Red, Thermo Fisher Scientific, Waltham, MA, USA) dissolved in DMSO to a final concentration of 1 μM, incubating for 1 min on the shaker at 20 rpm before washing twice with fresh medium, as described above. Likewise, we assessed the relationship with the tonoplast by administering the fluorescence auxin analogs to the NtTCP1-RFP line. For the competition experiments of the fluorescent auxin analogs with non-labelled auxins (NAA, IAA, and 2,4-D), the cells were incubated with 4 μL of 0.5 mM (2 μM) fluorescent auxin analogs. Then, 4 μL auxin stocks of 0.5 mM, 5 mM, or 25 mM (IAA and 2,4-D dissolved in 96% ethanol, NAA dissolved in 5 mM KOH), corresponding to 2, 20, or 100 μM of unlabeled auxin, were added. After 20 min of incubation, we washed the cells twice with fresh medium, as mentioned above. To determine the weighted co-localization coefficients of NBD-NAA with the ER, aliquots of 1 mL of 1-day-old WT BY-2 cells were first incubated with 4 μL of 0.5 mM NBD-NAA and 4 μL IAA (or 2,4-D) stocks of 0.5 mM or 25mM for 20 min on the shaker at 20 rpm. Then, we added 1 μL of 1 mM ER-Tracker for another 1 min of incubation, before washing the cells twice with fresh medium, as mentioned above. 

### 4.4. Generation of Protoplasts

The protocol was adapted from [70,71] with minor modifications as described in [72]. The source material were cells that had grown for 1 d after sub-cultivation. We digested aliquots of 4 mL cell suspension (non-transformed WT, NtTPC1A-GFP, or PIN1-GFP, respectively) harvested under sterile conditions at day 1 after sub-cultivation. The cells remained for 1 h at 26 °C in 4 mL enzyme solution (1% (*w*/*v*) cellulase YC (Yakuruto, Tokyo, Japan), and 0.1% (*w*/*v*) pectolyase Y-23 (Yakuruto, Tokyo) in 0.4 mol/L mannitol at pH 5.5) under constant shaking on a KS260 basic orbital shaker at 100 rpm in petri dishes of 90 mm diameter. After digestion, we collected the protoplasts, using a mild centrifugation at 500 rpm for 5 min in fresh reaction tubes. The protoplast sediment was carefully re-suspended in 10 mL of FMS wash medium containing 4.3 g/L Murashige and Skoog salts, 100 mg/L (myo)-inositol, 0.5 mg/L nicotinic acid, 0.5 mg/L pyroxidine-HCl, 0.1 mg/L thiamin, and 10 g/L sucrose in 0.25 M mannitol [70 Kuss-Wymer]. After three washing steps, we transferred the protoplasts into 4 mL FMS-store medium, (FMS wash medium) complemented with 0.1 mg/L 1-naphthalene-acetic acid (NAA, Sigma Aldrich, St. Louis, MO, USA) and 1 mg/L benzylamino-purine (BAP, Sigma Aldrich, St. Louis, MO, USA). Then, the protoplasts were ready for microscopical observation.

### 4.5. Acquisition of Microscopical Images

For the colocalization study, we used an AxioImager Z.1 microscope (Zeiss, Jena, Germany) equipped with an ApoTome microscope slider for optical sectioning and a cooled digital CCD camera (AxioCamMRm; Zeiss), recording GFP fluorescence from fluorescent auxin analogs through the filter set 38 HE (excitation at 470 nm, beamsplitter at 495 nm and emission at 525 nm). To resolve the cellular details of individual cells, and to score the co-localization of NBD-NAA and ER-Tracker, we used an AxioObserver Z.1 (Zeiss, Jena, Germany) inverted microscope equipped with a laser dual spinning disk scan head (Yokogawa CSU-X1 Spinning Disk Unit, Yokogawa Electric Corporation, Tokyo, Japan). A cooled digital CCD camera (AxioCamMRm; Zeiss) recorded the signals induced by the two laser lines (488 nm and 561 nm, Zeiss, Jena, Germany) attached to the spinning disk’s confocal scan head. We used a Plan-Apochromat 63×/1.44 DIC oil objective and operated image acquisition via the ZEN 2012 (Blue edition) software platform. Confocal z-stacks consisting of 10–30 individual sections were collected, and orthogonal projections were generated using the maximal intensity algorithm.

### 4.6. Co-Localization Analysis

The protocol was adapted from the protocol provided by Zeiss (Jena, Germany) with minor modifications using the “Co-localization” function to analyze two-channel images. To calibrate the cross-correlation scatterplot for the samples labeled for both NBD-NAA and ER-Tracker Red, the control sample images labeled for only one of the markers (either NBD-NAA or ER-Tracker Red, respectively) were required. First, we imaged the double-labeled samples with NBD-NAA and the ER-Tracker through the two fluorescent channels. Subsequently, we imaged the two control samples with exactly the same microscope settings as those for the double-labeled samples. After selecting the region of interest and examining the respective scatterplots for the sample of the NBD-NAA labeled control sample, we constructed the intensity histogram of the NBD-NAA signal in the double-labelled sample. The upper limit of this histogram served then to determine the so-called horizontal crosshair. In the same manner, the control stained by the ER-Tracker alone allowed us to determine the so-called vertical crosshair for the ER-signal in the double-labeled sample. For the analysis of the double-labeled (NBD-NAA and ER-Tracker) samples, the coordinates of these cross hairs must remain constant. Based on this calibration, the software can then calculate the weighted co-localization coefficients for each channel. These weighted co-localization coefficients are the ratio of co-localized pixels in the region of interest, based on the fluorescence intensity of each individual pixel.

### 4.7. Quantitative Image Analysis

To measure the fluorescence intensity in a selected region, we used the ImageJ freeware (https://imagej.net), imaging cells after incubation with NBD-NAA using an AxioImager Z.1 microscope. After selecting the regions of interest using the drawing/selection tools, we measured the parameters “area”, “integrated density”, and “mean gray value”. A region from the neighborhood of the cell served as a reference to determine the background signal. Based on the measurement for the entire cell population from one sample, the corrected total cell fluorescence (CTCF) was then determined as:CTCF = ID − A_c_ × I_b_,
where ID is the integrated density, A_c_ is the cross-area of the selected cell, and I_b_ is the mean background fluorescence. The Corrected Single Cell Fluorescence (CSCF) was then calculated as the ratio of CTCF over the number of scored cells.

## 5. Conclusions

Auxin is a small molecule, but it can deploy a plethora of different responses. Our work was motivated by the question: where does specificity come from? One possibility is that of the different receptors that are localized differently and, thus, can deploy differential signaling chains. To address this, we used fluorescently labelled analogs of auxin that are physiologically active but are transported similarly to their non-fluorescent counterparts, such that their localization is expected to reflect the localization of their non-fluorescent counterparts. We find that NBD-IAA, the analog of natural auxin, binds to both the ER and the tonoplast, while NBD-NAA, the analog of NAA, an artificial auxin that preferentially activates cell expansion rather than cell division [31], binds to the tonoplast only. Since the binding site at the ER and that on the tonoplast show differential ligand-binding patterns, they are likely to represent different molecules. 

Of course, the limitations of our model have to be considered. Although the differential localization of NBD-NAA (tonoplast) and NBD-IAA (ER) is significant (Figure 2), there seem to be minor pools that appear to locate ectopically from the bulk population. Although apparently minor in abundance, it would be interesting to conduct an experiment, where the cross-localization of the respective organelle signal versus the NBD-signal is quantified over a series of progressive out-competitions with non-labelled auxins. Furthermore, to what extent the situation in a suspension cell culture reflects the behavior in a tissue context remains to be elucidated. Since tobacco BY-2 recapitulates part of the developmental program of its source tissue, tobacco pith parenchyma [73], the model transfer might be feasible. To test this, we launched work with rice roots to follow the behavior of the NBD-auxins during gravitropic curvature, which is brought about by cell expansion in the distal elongation zone.

Our data support a concept, where auxin, through its interaction with different binding sites, can deploy different signaling. The specificity for the auxin signal would be, thus, not part of the molecule but embodied in the spatial pattern of the signal recipient, which is in line with the Organon model for communication elaborated by Karl Bühler in 1934 [74].

## Figures and Tables

**Figure 1 ijms-23-08593-f001:**
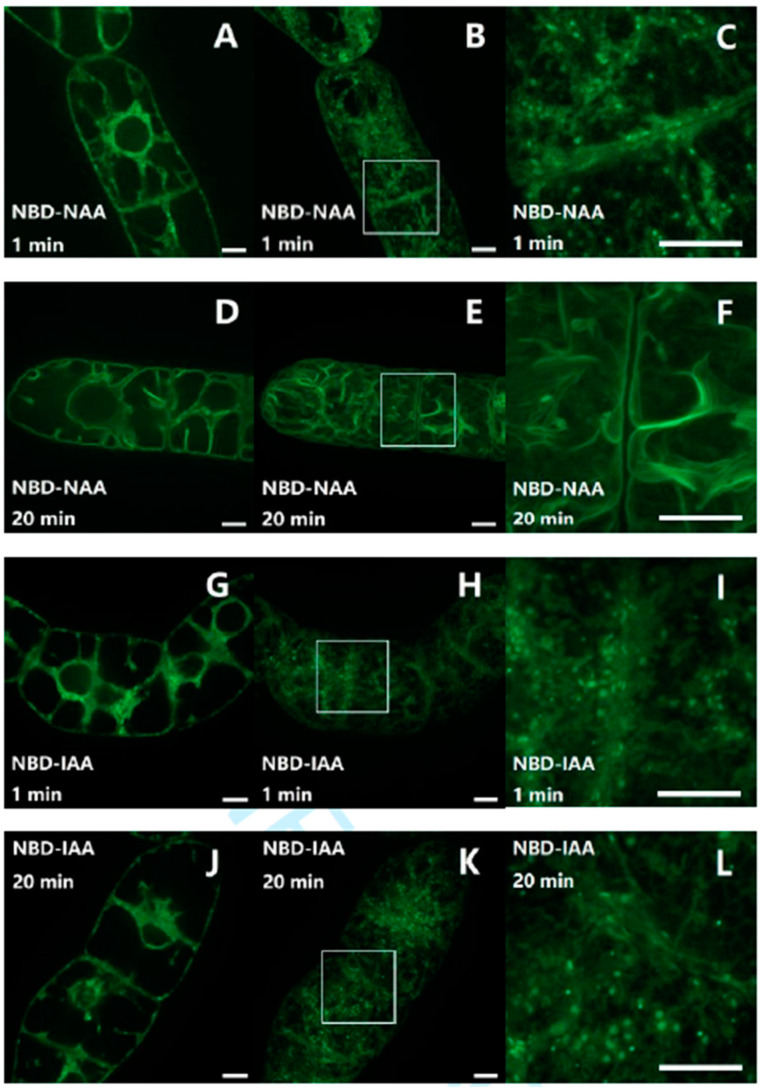
Subcellular localization of NBD-NAA (**A**–**F**) and NBD-IAA (**G**–**L**) in non-transformed BY-2 cells, recorded either after 1 min (**A**–**C**,**G**,**H**) or after 20 min (**D**–**F**,**J**–**L**) of incubation. Confocal sections in the central region (**A**,**D**,**G**,**J**), and geometric projections of the z-stack (**B**,**E**,**H**,**K**) are shown. (**C**,**F**,**I**,**L**) show the details of the region highlighted in (**B**,**E**,**H**,**K**). Both auxin analogs were administered in 2 μM. Scale bars represent 10 μm.

**Figure 2 ijms-23-08593-f002:**
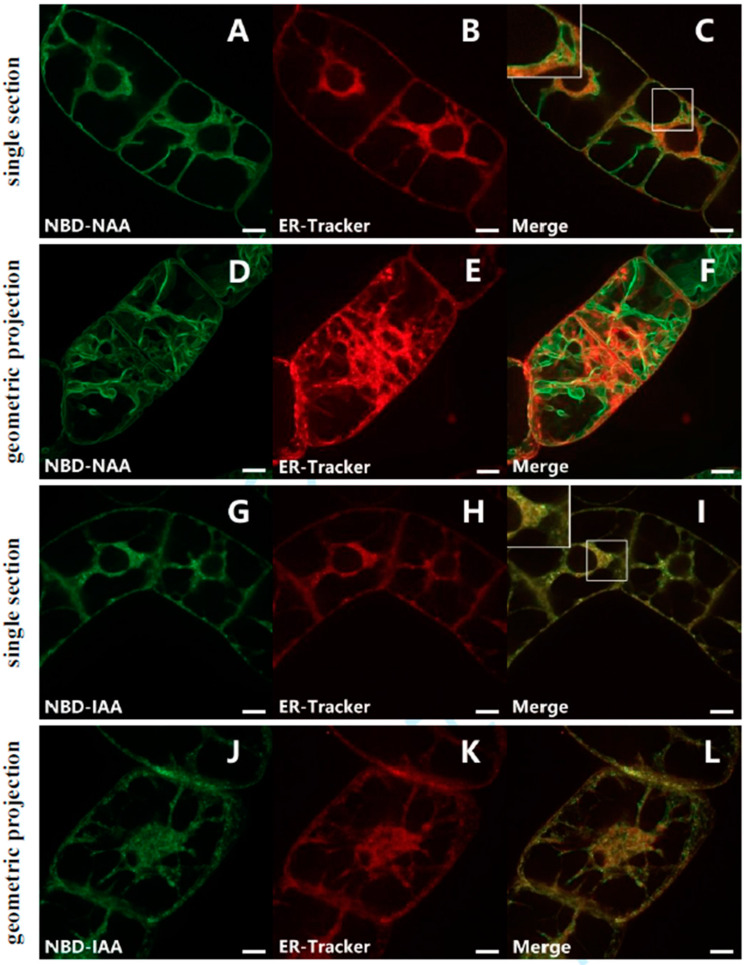
Subcellular localization of NBD-NAA (**A**–**F**) and NBD-IAA (**G**–**L**) in non-transformed BY-2 cells. The cells were pre-incubated with 2 μM of the fluorescent auxin analogs for 20 min and then incubated with 1 μM ER-Tracker Red for 1 min. Images of NBD-NAA (**A**,**D**) or NBD-IAA (**G**,**J**) and ER-Tracker Red (**B**,**E**,**H**,**K**) were merged (**C**,**F**,**I**,**L**) to show the relative localization of the two signals. For each analog, there are confocal sections in the central region.

**Figure 3 ijms-23-08593-f003:**
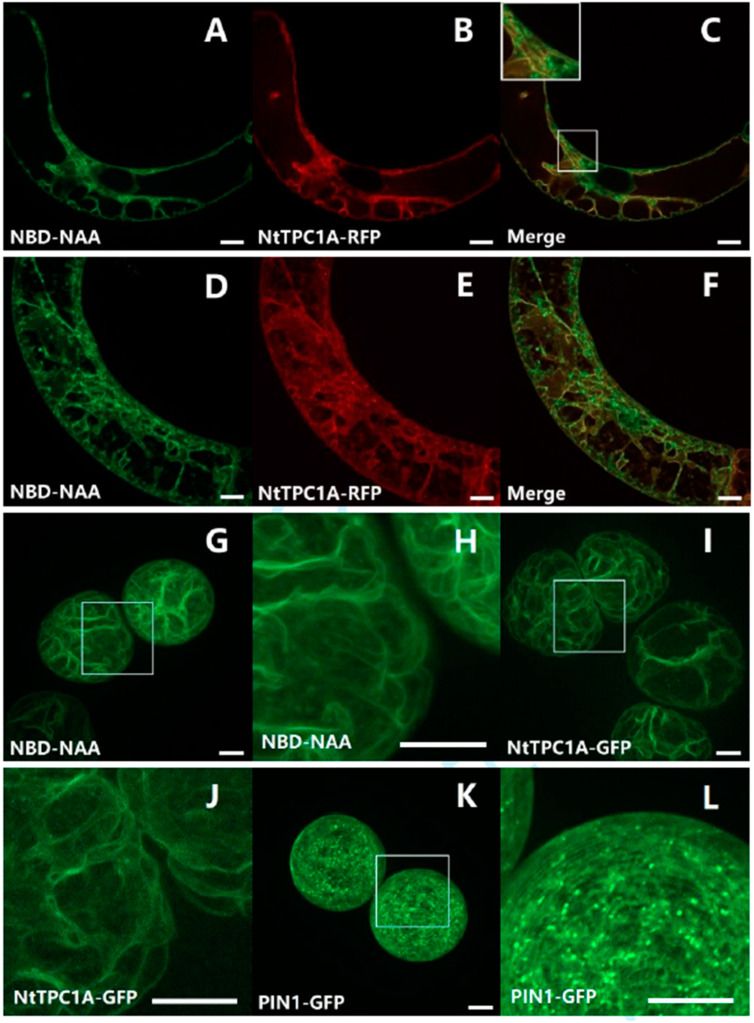
Evidence for a tonoplast localization of NBD-NAA. (**A**–**F**) Dual localization of NBD-NAA (**A**,**D**) with the tonoplast marker NtTPC1A-RFP (**B**,**E**) and a merge of both signals (**C**,**F**). The tonoplast marker was introduced by transient Agrobacterium-mediated transformation, and 2 μM NBD-NAA were administered for 20 min. (**G**–**L**) Comparison of the localization patterns for NBD-NAA (**G**,**H**), the tonoplast marker NtTPC1A-GFP (**I**,**J**) and the plasma membrane marker PIN1-GFP (**K**,**L**) in protoplasts generated from BY-2 cells. The details of the selected regions in (**G**,**I**,**K**) are shown in (**H**,**J**,**L**), respectively. Scale bars represent 10 μm.

**Figure 4 ijms-23-08593-f004:**
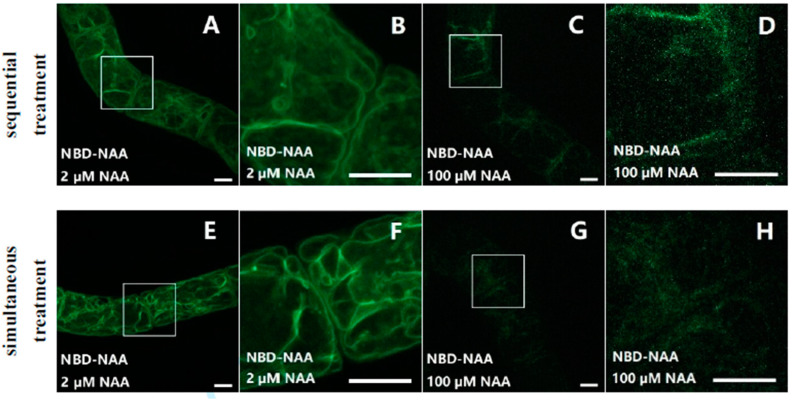
NBD-NAA and NAA compete for the same binding sites. Cells were incubated with NBD-NAA and different concentrations of NAA, either sequentially (**A**–**D**) or simultaneously (**E**–**H**). In both treatments, we administered the fluorescent analog for 20 min. In case of the pretreatment with non-fluorescent NAA, the treatment lasted for 20 min as well. Exemplarily, the results for 2 μM NAA (**A**,**B**,**E**,**F**), and for 100 μM NAA (**C**,**D**,**G**,**H**) are shown. To give the details of subcellular localization, the region marked by a white square in (**A**,**C**,**E**,**G**) is zoomed in on (**B**,**D**,**F**,**H**). Scale bar represents 10 μm.

**Figure 5 ijms-23-08593-f005:**
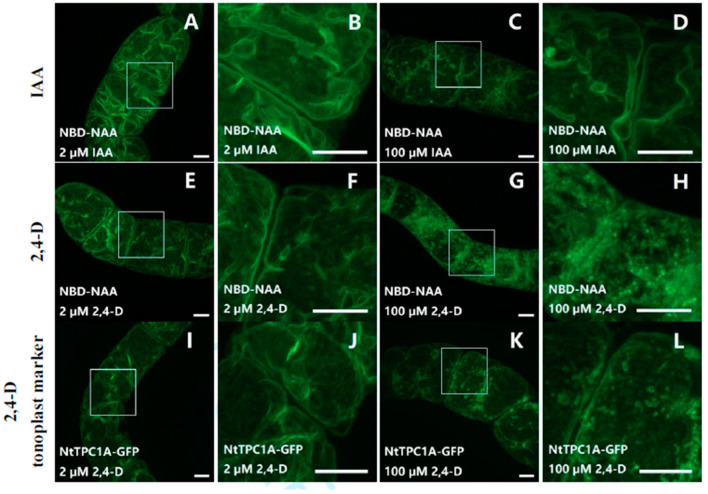
Neither IAA nor 2,4-D compete for the same binding sites as NBD-NAA (**A**–**H**). The cells were co-incubated with 2 μM NBD-NAA and either IAA (**A**–**D**) or with 2,4-D (**E**–**H**) for 20 min. The concentration of the non-labeled auxins was 2 μM (**A**,**B**) or (**E**,**F**), respectively) or 100 μM (**C**,**D**) or (**G**,**H**), respectively). The details of selected regions in (**A**,**C**,**E**,**G**) are shown in (**B**,**D**,**F**,**H**), respectively. To probe for the potential effect of 2,4-D on vacuolar organization, cells expressing the tonoplast marker NtTPC1A-GFP were treated with either 2 μM (**I**,**J**) or with 100 μM (**K**,**L**) of 2,4-D for 20 min. Scale bar represents 10 μm.

**Figure 6 ijms-23-08593-f006:**
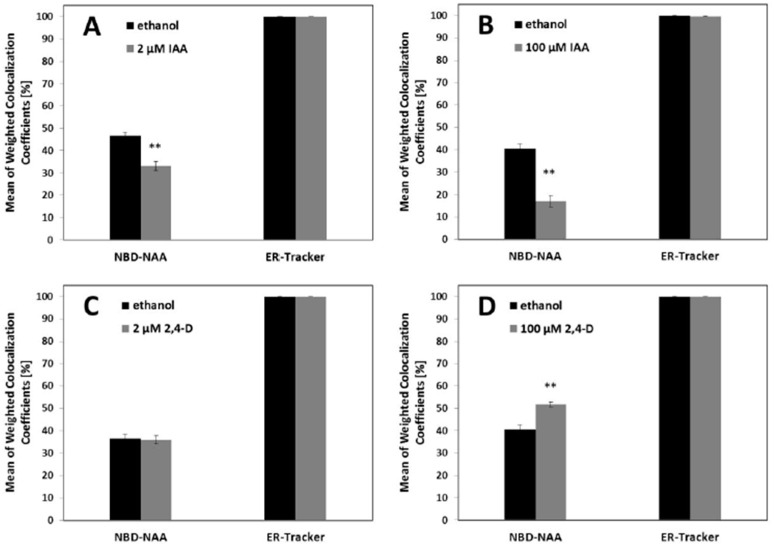
Mean values of weighted co-localization coefficients of NBD-NAA or ER-Tracker in non-transformed BY-2 cells. Cells were treated with either 2 μM NBD-NAA or the equivalent concentration of ethanol as solvent control before adding 1 μM ER tracker and incubating for 1 min. This experiment was conducted after pre-incubation for 20 min with either unlabeled IAA (**A**,**B**) or unlabeled 2,4-D (**C**,**D**) administered either in a concentration of 2 μM (**A**,**C**) or 100 μM (**B**,**D**), respectively. Data represent mean values and SE from 50 individual cells. Asterisks represent statistically significant differences (Student’s Independent two-sample *t*-test) with *p* < 0.01 (**), respectively.

**Figure 7 ijms-23-08593-f007:**
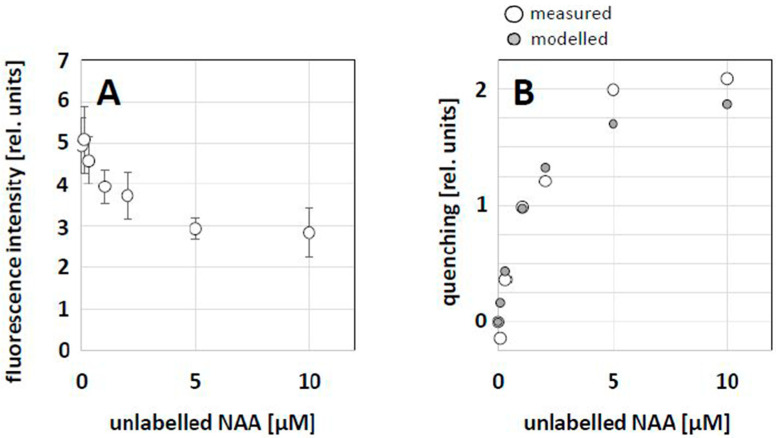
Competition between NBD-NAA and unlabeled NAA. Non-transformed BY-2 cells were treated with 2 μM NBD-NAA together with different concentrations of NAA for 20 min. (**A**) Relative fluorescence intensity of NBD-NAA over increasing concentrations of unlabeled NAA. Data represent mean and standard errors from three independent experimental series comprising a population of 900 individual cells. (**B**) Fitting of the observed fluorescence-quenching dependent on the concentration of unlabeled NAA using a Michaelis–Menten model assuming a K_D_ of 1.15 μM.

**Figure 8 ijms-23-08593-f008:**
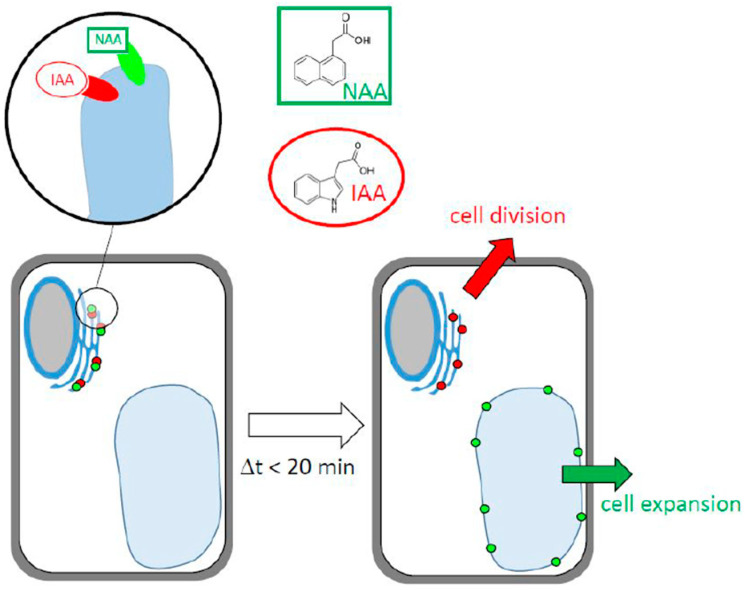
Working model explaining the data of the current study. Two auxin-binding sites with different affinities and ligand preferences localize to the ER. The site with a higher affinity for NAA translocates to the tonoplast and drives auxin-dependent events responsible for cell expansion, while the site with a higher affinity for IAA remains at the ER and might drive auxin-dependent events needed for cell division.

## Data Availability

All data supporting the findings of this study are available within the paper and within its Appendix A published online.

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
