# Peer review of "Fluorescent Auxin Analogs Report Two Auxin Binding Sites with Different Subcellular Distribution and Affinities: A Cue for Non-Transcriptional Auxin Signaling"

_ijms, 2022, doi:10.3390/ijms23158593_

Round 1

Reviewer 1 Report

In the manuscript entitled Fluorescent auxin analogs report two auxin binding sites with different subcellular distribution and affinities: a cue to non-transcriptional auxin signaling; Xiang Huang and collegues widen their former published researches regarding the development of fluorescent synthetic auxin analogs that allow to track the auxin movement and transport at the subcellular level.

They investigate the possibility that these fluorescent labelled molecules (NBD-NAA and NBD-IAA) could serve as proxies to delve into different subcellular perception sites for different type of auxins (IAA and NAA), which have been suggested to trigger diverse growth responses.

The NBD-auxins, according to the authors intentions, were originally designed and published as being able to mimic native auxin movements but not to be functional at the transcriptional-canonical-signaling (TIR1/AFB-IAAs mediated), and perchance not to feedback on auxin transport itself through that response.

By means of microscope imaging of tobacco protoplasts cells, the authors found that while initially both NDB-NAA and NBD-IAA initially localize at the endoplasmic reticulum, after 20 minutes only NBD-NAA localizes at the tonoplast.  According to the authors, this support the hypothesis -corroborated also by some previous reports- that there are differential sites of auxin perception within the cells. Those can, then, trigger non-canonical auxin signaling.

The topic and the question of the manuscript are intriguing: so far, many results have been accumulated which point to other and more neglected auxin mechanisms of action and perception. Though, the field is very controversial: the results are contrasting and  still need to be investigated more deeply.

In general, my main concern with this manuscript is that all the observations and the results were collected from Tobacco protoplasts, which might show, by their nature, very different physiological responses and behaviors from those observable in planta when treated with exogenous compounds. Moreover, NAA it’s not a natural occurring auxin, making hard to claim that physiologically there are in planta two distinct mechanisms of auxin perception. For these reasons, I would appreciate if the authors could discuss on the above concerns. 

Besides these concerns, I have some suggestions and I raise few issues that the authors need to revise in light of what follows:

-        I feel that the authors could make more convincing the subcellular localization of the NBD-auxins by adding second evidence (at least for the initial experiments in fig 1) and not by relying only on a single organelle-specific fluorescent dye/molecular marker.

-        I cannot find for most of the figures (the experiments) a formal colocalization analysis of any kind (like a Pearson Correlation Coefficient or similar). Are those missing? It would be worthy to perform them for clarity and add them.

-        From the results reported in figures 4 and 5, it is not clear to me if and why the binding of the NBD-IAA is reversible from its site of "perception" . Can the authors comment/expand more?

Minor points to amend:

-        No keywords have been listed.

-        The first time the acronyms appear in the text it would be helpful for the readers to have all the full names (almost no acronym is explicit in the manuscript).

-        Please, make explicit the concentration of the NBD-auxins (even if it is always 2 microM according what it is written in Mat and Methods section) throughout the main text (not always present) and make sure it is also in all the figures.

-        In the Intro section, add references for the DR5 and DII reporters in the sentences in lines 111-116.

-        Could the authors add in the Intro or in the discussion section some comments on the recently developed auxin biosensor (Herud-Sikimić, O., Stiel, A.C., Kolb, M.et al.A biosensor for the direct visualization of auxin.Nature 592,768–772 (2021). https://doi.org/10.1038/s41586-021-03425-2), which is genetically encoded and on-off switchable? I appreciate that this sensor and the fluorescent labelled auxin used in this study are conceptually very different, but still there are some margins of overlap among the kind of observations these two systems can allow at the sub cellular level. 

-        There are few typos throughout the ms that a common writing software can easily spot.

Reviewer 2 Report

The manuscript investigates the subcellular localization of auxin-binding sites of fluorescent auxin analogs. These analogs closely mimic NAA and IAA with respect to polar transport but are inactive for auxin signaling and metabolism. Authors showed that different auxin species NBD-NAA and NBD-IAA bind to other subcellular targets (ER and tonoplast). The manuscript indicates that NBD-NAA is localized in ER and tonoplast, and NBD-IAA binds only to the ER.

Major point:

I think that the conclusion of the proposed working model is too general. The authors haven’t shown an experiment to justify IAA localized at the endoplasmic reticulum controls cell division and that the NAA localized on the tonoplast is in charge of cell expansion.

Minor point:

In material and methods, I miss how numbers of Z-stack have been done to get pictures for geometric (orthogonal) projection.
